# Negative Sampling in Next-POI Recommendations: Observation, Approach, and Evaluation

## ABSTRACT

To recommend the *points of interest* (POIs) that a user would *check-in next*, most *deep-learning* (DL)-based existing studies have employed *random negative* (RN) sampling during model training. In this paper, we claim and validate that, as the training proceeds, such an RN sampling in reality performs as sampling *easy negative* (EN) POIs (*i.e.*, *EN sampling*) that a user was *highly unlikely* to check-in at her check-in time point. Furthermore, we verify that EN sampling is more *disadvantageous* in improving the accuracy than sampling *hard negative* (HN) POIs (*i.e.*, *HN sampling*) that a user was *highly likely* to check-in. To address this limitation, we present the novel concept of the *Degree of Positiveness* (DoP), which can be formulated by two factors: (i) the degree to which a POI has the *characteristics preferred* by a user; (ii) the *geographical distance* between a user and a POI. Then, we propose a new *model-training scheme based on HN sampling* by using DoP. Using real-world datasets (*i.e.*, NYC, TKY, and Brightkite), we demonstrate that all the state-of-the-art models trained by our scheme showed dramatic improvements in accuracy by up to about 82.8%. The code of our proposed scheme is available in an external link (https://anonymous.4open.science/r/code-BF64/).

**ACM Reference Format:**
Anonymous Author(s). 2023. Negative Sampling in Next-POI Recommendations: Observation, Approach, and Evaluation. In *Proceedings of Make sure to enter the correct conference title from your rights confirmation email (TheWebConf '24)*. ACM, New York, NY, USA, 11 pages. https://doi.org/XXXXXXX.XXXXXXX

## 1 INTRODUCTION

Recently, *location-based social network services* (LBSNs) (*e.g.*, Brightkite, Foursquare) have been widely used all over the world [4, 24]. On LBSN platforms, users *check-in points-of-interest* (POIs), *i.e.*, locations or places that they have visited. The vast amounts of such check-in records obtained from users triggered the intensive research on *next-POI recommender systems* [10, 22].

A user's *POI check-in sequence* indicates the *chronological order* for POIs that the user has checked-in along with their *check-in time points*. Next-POI recommendations are based on the intuition that recommender models trained by a user's POI check-in sequence will be able to infer her check-in pattern, thereby finding the POIs that the user would visit next time (*i.e.*, next-POIs).

With the growth of *deep-learning* (DL) technology, models that can learn user preferences for items from sequences of users' feedback effectively, such as RNN [17], LSTM [8], and Transformer [20], have been mainly employed for next-POI recommendations [12, 14]. DL-based next-POI recommender systems train models by determining a user's positive/negative POIs at every check-in time point of her POI check-in sequence. The POI that she checked-in at that point is regarded as *positive*, while the (remaining) POIs that she did not check-in are regarded as *negative*. We note that there are a large number of POIs not checked-in by a user at that time point; this makes a lot of room for design choices on which POIs should be selected as negative POIs for the positive POI. Since this design choice can significantly influence the recommendation accuracy, extensive research has been conducted on *negative sampling* in various recommendation domains such as OTT and e-commerce [16, 18, 26, 28, 29]. However, in the domain of next-POI recommendations, most approaches simply employ the *random sampling* on non-visited POIs for each user during model training [3, 13, 15, 23, 27]. In this paper, we aim to conduct the first *exhaustive and comprehensive study on negative sampling in the next-POI domain*.

First, we categorize the negative POIs into three groups: (i) POIs that a user was *highly likely* to check-in at that point but did not; (ii) POIs that a user was *highly unlikely* to check-in at that point; and (iii) POIs that belong to neither group (i) nor group (ii). POIs in group (i) are likely to have *characteristics* that she prefers at that point, making it *difficult* for the model to distinguish them from her corresponding positive POI at that point. In this sense, the POIs in group (i) can be regarded as *hard negative* (HN) POIs of the user at that point. In contrast, the POIs in group (ii) can be regarded as *easy negative* (EN) POIs since they are *easily* distinguished from the positive POI.

*Random negative* (RN) sampling employed in existing studies [3, 13, 15, 23, 27] is to sample negative POIs *without distinguishing* between HN POIs and EN POIs. In this paper, however, we claim that sampling HN POIs rather than EN POIs as negative POIs for the corresponding positive POI, *i.e.*, *HN sampling*, can be more effective in improving the accuracy of next-POI recommendations. By HN sampling, the models will be trained toward capturing precisely the characteristics of POIs that the user prefers more, which makes the ranking of the positive POI be predicted correctly (*i.e.*, higher than any other HN POIs of the user). To verify this claim in Section 2, we empirically investigate how the ranking of the positive POI predicted by the model can be *changed* depending on whether it is learned together with the user's HN POIs or EN POIs. Furthermore, by demonstrating that RN sampling behaves in reality as *EN sampling* as model training progresses, we validate why RN sampling will be *less effective* than HN sampling.

To address these limitations, we propose a new *model-training scheme based on HN sampling* for the next-POI recommendation. To this end, we define the *Degree of Positiveness* (DoP) that a user has

for a POI as the degree to which the user is likely to check-in the POI at a time point. A user's DoP for a POI can be determined by the following two factors at the time point: (i) the degree to which a POI has the *characteristics preferred* by the user, based on her previous check-in sequence up to the given point; (ii) the *geographical distance* between a user and a POI at the point, which is crucial in the next-POI recommendation. With these two factors, we formulate DoP, thereby determining the negative POIs with *high DoPs* as the HN POIs of the user. In detail, our proposed model-training scheme based on DoP is designed in two steps below.

**Step 1. Filtering POIs by the preferred characteristics.** Basically, the HN items in the general recommendation domain indicate the items *positioned very close* to the positive item in the *latent feature space* [16, 28]. In the next-POI recommendation, the distance of a negative POI from the positive POI in the latent space is likely to become *larger* as the negative POI has *fewer* characteristics preferred by a user based on her previous check-in sequence. Therefore, we *filter out* POIs with a *low degree* of having the preferred characteristics, *i.e.*, POIs *positioned far away* from the positive POI in the latent feature space, from the candidates for HN POIs.

**Step 2. Sampling HN POIs via DoP.** Unlike other recommendation domains such as OTT and e-commerce where considering only the distance in the latent feature space is *sufficient* to find the HN items, we should consider the *geographical distance* between POIs and a user as well in the next-POI domain; POIs *geographically located far away* from the user would be unlikely to be checked-in by her at that time point. Therefore, by Step 2, we aim to find the $n$ HN POIs among the POIs obtained by Step 1, by considering not only the degree of having the characteristics preferred by her but also the *geographical closeness* to the user at that point. As a result, we can effectively identify the user's HN POIs that are located close to the user's positive POI, *both in the latent feature space and in the geographical space*, through the concept of DoP.

Along with the $n$ HN POIs and the corresponding positive POI, we train a recommender model for the check-in point. With the model trained in this way for every user's all check-in points, we recommend the *next-POIs* that the user would check-in. We note that any existing models for the next-POI recommendation (*e.g.*, PLSPL [23], STAN [15]) can be incorporated with our proposed scheme, *i.e.*, *our scheme is model-agnostic*. In Section 4, we empirically show that state-of-the-art models benefit *significantly* with respect to accuracy from being trained by our scheme.

To the best of our knowledge, this is the first work to construct an *in-depth* study for the negative sampling issue in the next-POI recommendation that considers domain characteristics carefully. Our main contributions are as follows:

- *Key Observation*: (1) we find that RN sampling employed in existing studies behaves in reality as EN sampling does as model training proceeds; (2) we exhibit that HN sampling is beneficial to providing more-accurate next-POI recommendations than EN sampling.
- *Novel Approach*: we formulate DoP, a new measure to determine HN POIs, which considers both user preference and geographical distance, and we propose a model-training scheme based on HN sampling for accurate next-POI recommendations.

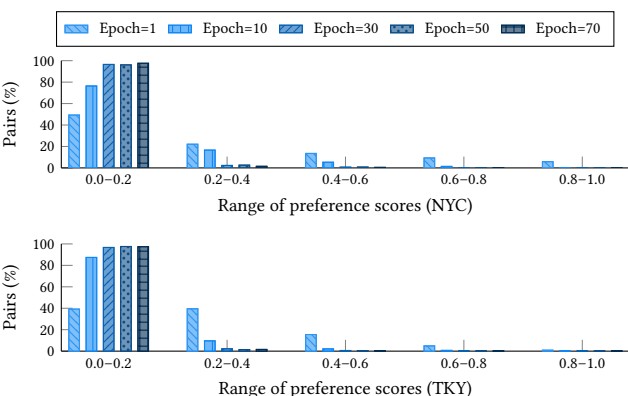

**Figure 1: Percentage of user-POI pairs according to the range of preference scores predicted by GeoSAN for each epoch.**

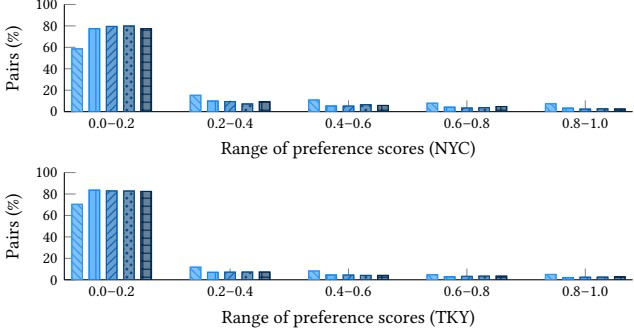

**Figure 2: Percentage of user-POI pairs according to the range of preference scores predicted by CatDM for each epoch.**

- *Extensive Evaluation*: we demonstrate that our proposed scheme can significantly enhance the accuracy of the models by using three real-world datasets and five state-of-the-art next-POI recommendation models.

## 2 MOTIVATION

In this section, first, we examine RN sampling methods employed in existing studies for the next-POI recommendation and highlight that they are close to EN sampling. Then, we show their limitations in terms of ranking prediction.

### 2.1 Negative sampling in existing studies

Negative POIs, which have not been checked-in by a user at a time point, can be grouped as follows, depending on *how probable* she would have checked-in at that point: (i) *hard negative* (HN) POIs with *high* probability; (ii) *easy negative* (EN) POIs with *low* probability; and (iii) the others. However, most existing studies for the next-POI recommendation have *not considered* these various types of negative POIs in model training. CatDM [27], STAN [15], and STKGRec [3] *randomly sample* negative POIs among *all* POIs that were not checked-in by a user. In addition, GeoSAN [13] and

TGSTAN [2] *randomly* sample negative POIs among the POIs that were (i) *geographically close* to a user but (ii) were not checked-in.

These models are trained in such a way that negative POIs are predicted to have *a low preference score*. Note that, as training proceeds (*i.e.*, the number of epochs increases), more non-checked-in POIs are sampled as negative POIs. That is, the number of POIs predicted to have a *low preference score* by the model (*i.e.*, the POIs regarded as *EN POIs*) gradually increases as training proceeds. Thus, we claim that RN sampling employed in existing studies performs similarly as *EN sampling* does as training proceeds.

To verify this claim, we empirically investigate how the respective ratio of HN/EN POIs among the negative POIs of a user varies as the model is trained, by using two real-world datasets (*i.e.*, NYC and TKY) [25].[1] First, for each user, we select $M$ POIs that she has not checked-in at each time point in the sequence.[2] Then, we predict her preference scores for these $M$ POIs from 0 to 1 by the recommender model (which is being trained). We repeat this process in training the model for a total of 70 epochs, and we split all pairs of user-POI into five groups based on their preference scores (*i.e.*, 0.0–0.2, 0.2–0.4, ... and 0.8–1.0). As models, we employ GeoSAN [13] and CatDM [27], which are state-of-the-art methods for the next-POI recommendation.

Figures 1 and 2 show the results for GeoSAN and CatDM, respectively. The $x$-axis denotes the range of preference scores predicted by the model and the $y$-axis denotes the percentage of user-POI pairs corresponding to each range. The POIs belonging to the range of 0.0–0.2 and the range of 0.8–1.0 can be regarded as EN POIs and HN POIs of the user, respectively. From the middle phase of the training (*i.e.*, the 30th epoch), we observe that most pairs (more than 95% and 80% by GeoSAN and CatDM, respectively, for both datasets) belong to the range of 0.0–0.2; if a negative POI is randomly sampled for a user, it is *highly likely* to be an *EN POI* of that user. In this sense, the model trained with RN sampling is equivalent (in practice) to the model trained with *EN sampling* as the training progresses. In the following subsection, we will exhibit the limitations of the training scheme with this EN sampling.

## 2.2 Limitations of EN sampling

While employing EN sampling, the model is trained to predict a ranking for positive POIs higher than that for EN POIs. Here, note that the rankings of EN POIs are at the *bottom* among the total negative POIs for that user at that time point. Therefore, the ranking of the positive POI, which is predicted to be *only higher than that of EN POIs*, is likely to be located around the *middle*. That is, there may still remain many negative POIs with a *higher ranking than that of the positive POI*, which indicates that the model is being trained in the *incorrect way*.

To validate this claim empirically, we present the difference in the predicted rankings for positive POIs for the two models trained by HN sampling and EN sampling, respectively, by using two real-world datasets (*i.e.*, NYC and TKY). We employ GeoSAN [13] as the model and refer to GeoSAN-HN and GeoSAN-EN as the models

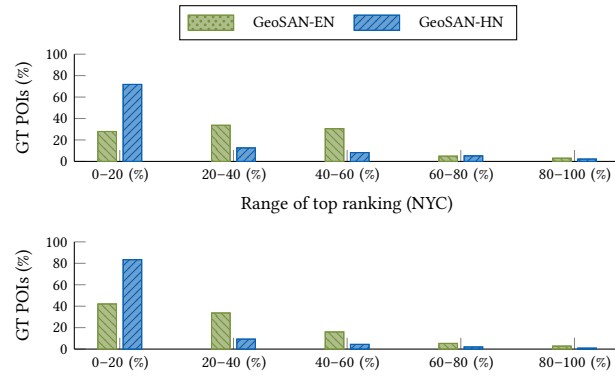

**Figure 3: Percentage of ground truth (GT) POIs according to the range of top ranking predicted by GeoSAN-EN and GeoSAN-HN, respectively.**

trained by HN sampling and EN sampling, respectively. GeoSAN-HN and GeoSAN-EN are trained by sampling the top-10 and bottom-10 POIs as negative POIs, respectively, based on the predicted preference scores among the total non-checked-in POIs at each time point. Then, through each trained model, we predict the preference score for the POI that a user checked-in *last* in the sequence (*i.e.*, *ground truth*, GT) and the preference scores for all negative POIs at that time point. After sorting all these POIs in descending order by the predicted scores, we divide GT POIs into five ranges according to their rankings among all POIs, *i.e.*, top 0–20%, ..., and 80–100%.

Figure 3 shows the results, where the $x$-axis represents the ranking range and the $y$-axis represents the percentage of GT POIs corresponding to each range. For the NYC dataset, we observe that more than 70% of GT POIs belong to the range of top 0–20% based on the rankings predicted by GeoSAN-HN; the rankings for most GT POIs are *correctly* predicted to be at the top among all POIs. On the other hand, by GeoSAN-EN, only about 30% of GT POIs belong to the range of top 0–20%; the rankings for the remaining (*i.e.*, around 70% of) GT POIs are *incorrectly* predicted to be around or below the middle, which can lead to *degraded* recommendation accuracy. To address this limitation, we propose a novel training scheme for next-POI recommendation models based on HN sampling.

## 3 PROPOSED APPROACH

In this section, we first define the *Degree of Positiveness* (DoP) that indicates the degree to which a user is likely to check-in a POI at a time point. With the concept of DoP, we present our model-training scheme based on HN sampling for the next-POI recommendation.[3]

## 3.1 Degree of Positiveness (DoP)

For a user $u$ at a given time point $t$, the following two factors decide DoP for a POI $v$.

- $c(u, v, t)$: the degree to which POI $v$ has the *characteristics* preferred by user $u$, given the previous check-in sequence of $u$ up to the time point $(t-1)$.

---

[1]https://sites.google.com/site/yangdingqi/home/foursquare-dataset/
[2]Following [13], we sample $M$ negative POIs that were *geographically close* to a user at each time point ($M$=2,000).

[3]For the notations used in this paper, please refer to Appendix B.

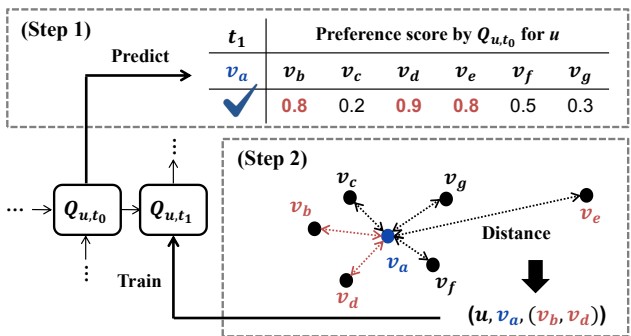

**Figure 4: Overview of our proposed model-training scheme.**

- $d(u, v, t)$: the geographical distance between user $u$ and POI $v$ at the time point $t$.

A positive POI that $u$ has checked-in at $t$ can be regarded as the POI with the highest DoP for $u$ at $t$. In this sense, HN POIs of $u$ at $t$ are regarded as the POIs *not-checked-in* by $u$ at $t$ but with a fairly *high DoP* comparable to that of the positive POI.[4]

There can be three strategies to give *priority between the two factors for DoP* in finding the HN POIs of each user: (i) prioritizing $c(u, v, t)$ over $d(u, v, t)$; (ii) prioritizing $d(u, v, t)$ over $c(u, v, t)$; and (iii) giving equal priority to both of them. Among these strategies, we adopt strategy (i): (Step 1) we first decide the candidates for HN POIs based only on $c(u, v, t)$, and then (Step 2) we sample HN POIs by using a *fine-grained preference score* adjusted by both $c(u, v, t)$ and $d(u, v, t)$. We empirically demonstrate that the HN POIs obtained from this strategy *significantly* help a model *converge to higher accuracy* in early- or mid-stages of training compared to the other two strategies in Section 4.

The overall procedure of our model-training scheme based on the HN sampling is illustrated in Figure 4. We explain the details of Steps 1 and 2 in Figure 4 in subsections 3.2 and 3.3, respectively.

## 3.2 Filtering POIs (Step 1)

As shown in Figures 1 and 2, HN POIs typically account for only a *very small portion* ($\leq 3\%$) during model training. In order to select such a small number of HN POIs precisely, we aim to filter out the negative POIs unlikely to become the HN POIs of $u$ at $t$ among the total negative POIs in Step 1.

The HN POIs are *not easily distinguished* from the positive POI of $u$ at $t$ by the model during the training process. In other words, the HN POIs can be regarded as the POIs *positioned very close* to the positive POI in the *latent feature space*. The degree of having characteristics preferred by a user on a POI is *highly associated* with the distance of the positive POI and that POI in the latent feature space; the POIs with a high/low degree of having the characteristics preferred by $u$ are positioned close/far from her positive POI, respectively, in the latent feature space. For this reason, the POIs with a *low degree of having the characteristics* preferred by $u$ are *unlikely* to become the HN POIs of $u$ at $t$ (since the model will predict preference scores for them *much lower* than that of the positive POI

---

[4]Following the existing studies [3, 27], we exclude the POIs that have been checked-in by $u$ before $t$ from the negative POIs at $t$.

of $u$ at $t$). To filter our such POIs, we consider $c(u, \bar{v}, t)$ and leverage the model trained by the $u$'s check-in sequence up to $(t-1)$ (*i.e.*, the model that is being trained for the next-POI recommendation) to determine $c(u, \bar{v}, t)$.

Here, any existing models that can infer the user's check-in pattern from the check-in sequence (*e.g.*, GeoSAN [13], STAN [15]) can be employed. We formulate $c(u, \bar{v}, t)$ with the preference score of $u$ for $\bar{v}$ predicted by the model as follows:

$$c(u, \bar{v}, t) = Q_{u,(t-1)}(\bar{v}), \quad (1)$$

where $Q_{u,(t-1)}$ denotes the model trained by the check-in sequence of $u$ up to $(t-1)$; $Q_{u,(t-1)}(\bar{v})$ denotes the preference score of $u$ for $\bar{v}$ predicted by model $Q_{u,(t-1)}$ ($0 \leq Q_{u,(t-1)}(\bar{v}) \leq 1$).

Then, we sort the negative POIs of $u$ at $t$ in descending order of $c(u, \bar{v}, t)$ and filter out bottom-ranked POIs. The remaining negative POIs are defined as the set $C_{u,t}(m)$ of candidates for the HN POIs:

$$C_{u,t}(m) = \{\bar{v} | rank(c(u, \bar{v}, t)) \leq m\}, \bar{v} \in \bar{V}_{u,t}, \quad (2)$$

where $\bar{V}_{u,t}$ represents the set of all POIs that $u$ has not checked-in at $t$; $rank(c(u, \bar{v}, t))$ represents the ranking of $\bar{v}$ among the POIs in $\bar{V}_{u,t}$ after sorting; $m$ represents the size of $C_{u,t}(m)$. In other words, by Step 1, we select the top-$m$ POIs with the highest $c(u, \bar{v}, t)$ predicted by model $Q_{u,(t-1)}$ among the POIs in $\bar{V}_{u,t}$, and use them as candidates for the HN POIs of $u$ at $t$.[5]

For example, in Step 1 of Figure 4, the POI $v_a$ checked-in by a user $u$ is regarded as the positive POI at the time point $t_1$. Given $m=3$, among the POIs not checked-in by $u$ (*i.e.*, $v_b \sim v_g$), the top-3 POIs with the highest preference scores predicted by model $Q_{u,t_0}$ are determined as the candidates for the HN POIs at $t_1$ (*i.e.*, $C_{u,t_1}(3) = \{v_b, v_d, v_e\}$).

## 3.3 Sampling HN POIs via DoP (Step 2)

Unlike other recommendation domains, such as OTT and e-commerce, where considering only the distance in the latent feature space is *sufficient* to find the HN items [16, 18, 28], we claim that the *geographical distance* should be also considered to find the HN POIs in the next-POI domain; if a negative POI $\bar{v}$ in $C_{u,t}(m)$ with a *high degree* of having the characteristics preferred $u$ at $t$ is geographically located *far away* from $u$, then $u$ is unlikely to check-in $\bar{v}$ at $t$.

Using real-world datasets, we conducted a statistical analysis to verify the relationship between a user's check-in probability for a POI and the distance between the POI and the user. Specifically, we calculated the geographical distance between two (positive) POIs that $u$ has *successively* checked-in on her check-in sequence. Then, we obtained the distribution of a distance between every pair of successive POIs.

Figure 5 illustrates the results for the NYC dataset,[6] where the $x$-axis indicates the range of a distance and the $y$-axis indicates the ratio of pairs of successive POIs corresponding to the range of a distance. It can be observed that about 70% of all pairs belong to the range of a distance less than 5km; this indicates that users generally

---

[5]To prevent $C_{u,t}(m)$ from always consisting of the same (top-$m$) POIs, we construct $\bar{V}_{u,t}$ of $M$ POIs randomly sampled from all negative POIs of $u$ at $t$ (*e.g.*, $M$=2,000), following the previous studies for HN sampling in other domains (*e.g.*, OTT, e-commerce) [18, 28].

[6]For the results of using another dataset, please refer to Appendix C.

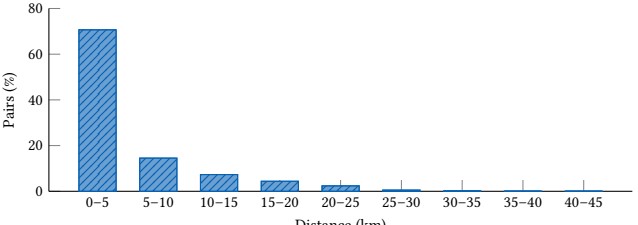

**Figure 5: Distribution of pairs of successive POIs over the distance between them (NYC).**

have a strong tendency to check-in the POI *located close to their current locations*.

Thus, in Step 2, we first obtain the distance between each POI $\bar{v} \in C_{u,t}(m)$ and $u$ at $t$. Specifically, we formulate the distance $d(u, \bar{v}, t)$ as follows, while regarding the location of the positive POI that $u$ has checked-in at $t$ as the location of $u$ at $t$:

$$d(u, \bar{v}, t) = Haversine(v_{u,t}, \bar{v}), \bar{v} \in C_{u,t}(m), \tag{3}$$

where $Haversine(\cdot)$ denotes the *Haversine formula* that determines the shortest distance between two locations by using their latitudes and longitudes [19], which has been widely employed in existing studies for the next-POI recommendation [9, 21]; $v_{u,t}$ denotes the positive POI of $u$ at $t$. Then, we perform the *min-max scaling* on the values of $d(u, \bar{v}, t)$ obtained for every POI $\bar{v} \in C_{u,t}(m)$: $d(u, \bar{v}, t)$ of $\bar{v}$ located farthest (resp. closest) from $u$ becomes 1 (resp. 0) (*i.e.*, $0 \leq d(u, \bar{v}, t) \leq 1$).

Then, we aim to determine the HN POIs among the POIs $\in C_{u,t}(m)$ by considering not only the degree of having the characteristics preferred by $u$ but also the geographical closeness to $u$ at $t$. Consequently, we formulate DoP of $u$ for $\bar{v}$ at $t$ by two factors (*i.e.*, $c(u, \bar{v}, t)$ and $d(u, \bar{v}, t)$) as follows:

$$DoP(u, \bar{v}, t) = (1 - d(u, \bar{v}, t)) \cdot c(u, \bar{v}, t), \tag{4}$$

where $DoP(u, \bar{v}, t)$ can be regarded as a *fine-grained preference score adjusted* by both $c(u, \bar{v}, t)$ and $d(u, \bar{v}, t)$. In other words, $DoP(u, \bar{v}, t)$ increases as $c(u, \bar{v}, t)$ increases or $d(u, \bar{v}, t)$ decreases. Note that *multiplication* adopted in Eq. 4 has been widely used in other studies for recommender systems [1, 5].[7]

Finally, we sample the top-$n$ HN POIs with the highest $DoP(u, \bar{v}, t)$ among the $m$ candidate POIs (*i.e.*, $C_{u,t}(m)$). We define the set $H_{u,t}(n)$ of these top-$n$ HN POIs of $u$ at $t$ as follows:

$$H_{u,t}(n) = \{\bar{v} | rank(DoP(u, \bar{v}, t)) \leq n\}, \tag{5}$$

where $n$ represents the number of HN POIs to sample. In other words, our proposed concept of DoP aids in effectively sampling $u$'s HN POIs which are located close to $u$'s positive POI *both in the latent feature space and in the geographical space*.

In Step 2 of Figure 4, while the preference scores predicted by $Q_{u,t_0}$ for $v_b$, $v_d$, and $v_e$ are similar to each other, $v_e$ is located farther from $v_a$ (*i.e.*, location of $u$) than the other two POIs. That is, $DoP(u, v_e, t_1)$ is smaller than $DoP(u, v_b, t_1)$ and $DoP(u, v_d, t_1)$. For this reason, given $n$=2, we can finally sample $v_b$ and $v_d$ as the HN POIs of $u$ at $t_1$ (*i.e.*, $H_{u,t_1}(2) = \{v_b, v_d\}$).

---

[7]Other methods to combine the two factors will be addressed in our further study.

### 3.4 Training a next-POI recommender model

Given the set of $H_{u,t}(n)$ of $n$ HN POIs of $u$ at $t$ obtained by Step 2 and the corresponding positive POI $v_{u,t}$, we train model $Q_{u,(t-1)}$ with the following *cross-entropy* loss [11]: [8]

$$\mathcal{L}_{u,t} = -(log(Q_{u,(t-1)}(v_{u,t})) +$$
$$\sum_{\bar{v} \in H_{u,t}(n)} log(1 - Q_{u,(t-1)}(\bar{v}))). \tag{6}$$

To minimize $\mathcal{L}_{u,t}$, $Q_{u,(t-1)}$ is trained toward predicting the preference score for $v_{u,t}$ close to 1 and the preference scores for $\bar{v} \in H_{u,t}(n)$ close to 0. We refer to $Q_{u,t}$ as the model trained by these positive POI and HN POIs of $u$ at $t$ in this way (*i.e.*, the model trained by the check-in sequence up to $t$).

At the "next" time point ($t$+1), we repeat Steps 1 and 2 through the model $Q_{u,t}$. For example, as displayed in Figure 4, we sample new HN POIs of $u$ at the next time point $t_2$ by using $Q_{u,t_1}$ which has been trained by $u$'s positive POI $v_a$ and two HN POIs $v_b$ and $v_d$ at $t_1$.

We repeat this process until the "last" time point in $u$'s check-in sequence. We also train the model $Q$ with respect to other users by repeating Steps 1 and 2 at every time point until the last point of their check-in sequence. Finally, through the model $Q$ trained by *full* check-in sequences of all users in this way, we find and recommend next-POIs for each user $u$ at the current time point.

### 3.5 Discussions

It is worth noting that our proposed model-training scheme *differs* from the previous HN sampling methods proposed for the general recommendation domains (*e.g.*, OTT, e-commerce) [16, 18, 26, 28, 29] in the following ways: (i) our proposed scheme aims to find the HN POIs of each user by considering not only the *distance between POIs in the latent feature space* but also their *geographical distance*, and (ii) it is necessary to consider a *sequence* of POIs a user has checked-in to infer user preferences in the next-POI recommendation. Our work contributes to providing an important insight that HN sampling can be more effective for *sequence-based recommender systems* than RN sampling.

Furthermore, by pointing out the limitations of the existing next-POI recommendation studies based on RN sampling, we propose a promising path for guiding future studies in this domain toward effective negative POI sampling, which is an important contribution in the field of data science.

## 4 EVALUATION

### 4.1 Experimental Setup

We performed experiments on three real-world datasets, which have been widely adopted in the next-POI recommendation: NYC and TKY from Foursquare [25] and Brightkite [4] as shown in Table 2.[9] We employed two popular metrics used in existing studies [13, 15, 27] to compute the accuracy: *hit rate* (namely, H) and *NDCG* (namely, G). As a (base) next-POI recommender model, we used

---

[8]In the case of employing models originally trained through a *pair-wise loss* (*e.g.*, CatDM [27], STKGRec [3]) as a recommender model, we exploit the pair-wise loss instead of the cross-entropy loss.
[9]The sparsity of the dataset equals the ratio of missing cells out of the total cells in the user-POI check-in matrix.

**Table 1: Comparison of accuracy between the original RN sampling scheme (*i.e.*, Orig) and our proposed scheme based on HN sampling (*i.e.*, Ours) for all base models**

| Dataset | Metric | PLSPL | | | CatDM | | | STAN | | | STKGRec | | | GeoSAN | | |
|---|---|---|---|---|---|---|---|---|---|---|---|---|---|---|---|---|
| | | Orig | Ours | Gain (%) | Orig | Ours | Gain (%) | Orig | Ours | Gain (%) | Orig | Ours | Gain (%) | Orig | Ours | Gain (%) |
| NYC | H@5 | 0.272 | **0.400** | 47.1 | 0.220 | **0.249** | 13.2 | 0.307 | **0.399** | 30.0 | 0.402 | **0.436** | 8.5 | 0.356 | **0.441** | 23.9 |
| | H@10 | 0.402 | **0.519** | 29.1 | 0.261 | **0.291** | 11.5 | 0.393 | **0.461** | 17.3 | 0.484 | **0.523** | 8.1 | 0.463 | **0.566** | 22.2 |
| | G@5 | 0.172 | **0.281** | 63.4 | 0.189 | **0.216** | 14.3 | 0.215 | **0.290** | 34.9 | 0.299 | **0.324** | 8.4 | 0.223 | **0.283** | 26.9 |
| | G@10 | 0.214 | **0.320** | 49.5 | 0.203 | **0.230** | 13.3 | 0.243 | **0.310** | 27.6 | 0.326 | **0.353** | 8.3 | 0.259 | **0.327** | 26.3 |
| TKY | H@5 | 0.172 | **0.314** | 82.5 | 0.188 | **0.227** | 20.9 | 0.209 | **0.310** | 48.3 | 0.398 | **0.427** | 7.3 | 0.534 | **0.641** | 19.9 |
| | H@10 | 0.257 | **0.433** | 68.5 | 0.241 | **0.282** | 16.8 | 0.290 | **0.387** | 33.4 | 0.469 | **0.503** | 7.2 | 0.630 | **0.743** | 18.0 |
| | G@5 | 0.115 | **0.210** | 82.8 | 0.164 | **0.200** | 22.2 | 0.142 | **0.220** | 54.9 | 0.306 | **0.332** | 8.5 | 0.385 | **0.460** | 19.7 |
| | G@10 | 0.142 | **0.248** | 74.6 | 0.182 | **0.218** | 19.9 | 0.168 | **0.245** | 45.8 | 0.329 | **0.356** | 8.2 | 0.419 | **0.500** | 19.1 |
| Bright -kite | H@5 | 0.655 | **0.737** | 12.5 | 0.633 | **0.640** | 1.2 | 0.569 | **0.699** | 22.8 | 0.68 | **0.696** | 2.4 | 0.650 | **0.668** | 2.7 |
| | H@10 | 0.707 | **0.774** | 9.5 | 0.644 | **0.652** | 1.3 | 0.644 | **0.753** | 16.9 | 0.736 | **0.754** | 2.4 | 0.724 | **0.789** | 9.1 |
| | G@5 | 0.470 | **0.609** | 29.5 | 0.626 | **0.634** | 1.3 | 0.433 | **0.570** | 31.6 | 0.568 | **0.585** | 3.0 | 0.390 | **0.474** | 21.7 |
| | G@10 | 0.488 | **0.621** | 27.3 | 0.630 | **0.638** | 1.3 | 0.469 | **0.588** | 25.4 | 0.587 | **0.604** | 3.0 | 0.427 | **0.513** | 20.1 |

**Table 2: Statistics of three real-world datasets**

| Dataset | # of users | # of POIs | # of check-ins | Sparsity (%) |
|---|---|---|---|---|
| NYC | 1,083 | 8,434 | 171,493 | 99.47 |
| TKY | 2,293 | 12,740 | 482,118 | 99.52 |
| Brightkite | 1,866 | 11,698 | 704,673 | 99.84 |

five state-of-the-art models: PLSPL [23], CatDM [27], STAN [15], STKGRec [3], and GeoSAN [13]. We set the number $m$ of candidates for the HN POIs and the number $n$ of HN POIs to sample at each time point to 50 and 10, respectively.

To evaluate the accuracy of base models precisely, we pre-processed the three datasets, divided each dataset into training/validation/test sets, and tuned the hyperparameters by following the respective study that proposed each model. For the specifics of the evaluation protocol, refer to Appendix A.

## 4.2 Experimental Results

Our experiments are designed to answer the following four key research questions (RQs).

- **RQ 1.** How effective is our proposed training scheme for state-of-the-art base models?
- **RQ 2.** How effective is our strategy of filtering out POIs in Step 1?
- **RQ 3.** How effective is our strategy of considering the distance in Step 2?
- **RQ 4.** How does the accuracy vary depending on values of $m$ or $n$?

*4.2.1 RQ 1.* For each base model, we compare the accuracy between the model trained by our proposed scheme based on HN sampling (*i.e.*, Steps 1 and 2) and the model trained by its original training scheme. For example, the original STAN [15] is trained by randomly sampling $n$ negative POIs among all negative POIs

of a user, while the original GeoSAN [13] is trained by randomly sampling $n$ negative POIs among the negative POIs geographically close to a user. We set the value of $n$ to 10 for all the models except PLSPL [23].[10]

In Table 1, for *all* the datasets, we observe that *all* models trained by our proposed scheme (*i.e.*, Ours) consistently and universally outperform the original model (*i.e.*, Orig) in terms of *all* the metrics. Specifically, for NYC, Ours shows accuracy *significantly higher* than Orig by up to about 82.8%, 22.2%, 56.3%, 8.5%, and 26.9% for PLSPL, CatDM, STAN, STKGRec, and GeoSAN, respectively, where the gain is computed by (Ours−Orig)/Orig×100. For Brightkite, the gains are slightly smaller than those for the other two datasets. We attribute this to the fact that the average number of POIs checked-in by a user in Brightkite (*i.e.*, 18) is much smaller than in NYC (*i.e.*, 44) and TKY (*i.e.*, 60), which might make it relatively difficult for a model to give accurate preference scores to negative POIs. Nonetheless, the consistent results for all the datasets successfully validate that our proposed scheme using the concept of DoP is *more beneficial* to improving accuracy than the RN sampling scheme employed in the existing models.

*4.2.2 RQ 2.* We designed two variants of our scheme with respect to filtering out POIs in Step 1: (i) filtering out negative POIs with the high value of $d(u, \bar{v}, t)$ (*i.e.*, the POIs geographically far from $u$ at $t$) and then sampling HN POIs based on DoP among the remaining POIs (namely, Filt-$d$); and (ii) sampling HN POIs based on DoP *without filtering* out any negative POIs (namely, w/o Filt). Between two factors of DoP, Filt-$d$ prioritizes $d(u, \bar{v}, t)$ over $c(u, \bar{v}, t)$, while w/o Filt gives equal priority to two factors.

Table 3 shows the results on the NYC and TKY datasets. The order of the three schemes from the most accurate to the least accurate is (Ours > w/o Filt > Filt-$d$) regardless of the models, which indicates the following observation: *prioritizing the preference score*

---

[10]PLSPL [23] is trained by using *all* negative POIs of a user *without sampling*. To reduce the computation overhead, we trained the original PLSPL with a value of $n$ large enough ($n$=500) to show results similar to its original results.

**Table 3: Comparison of accuracy among our scheme and its two variants with respect to filtering out POIs in Step 1**

| Dataset | Metric | PLSPL | | | CatDM | | |
|---|---|---|---|---|---|---|---|
| | | Filt-$d$ | $w/o$ Filt | Ours | Filt-$d$ | $w/o$ Filt | Ours |
| NYC | H@5 | 0.328 | 0.381 | **0.400** | 0.226 | 0.234 | **0.249** |
| | H@10 | 0.453 | 0.508 | **0.519** | 0.267 | 0.282 | **0.291** |
| | G@5 | 0.220 | 0.261 | **0.281** | 0.195 | 0.202 | **0.216** |
| | G@10 | 0.260 | 0.302 | **0.320** | 0.209 | 0.218 | **0.230** |
| TKY | H@5 | 0.207 | 0.268 | **0.314** | 0.187 | 0.192 | **0.227** |
| | H@10 | 0.347 | 0.396 | **0.433** | 0.241 | 0.247 | **0.282** |
| | G@5 | 0.127 | 0.171 | **0.210** | 0.159 | 0.166 | **0.200** |
| | G@10 | 0.172 | 0.212 | **0.248** | 0.177 | 0.184 | **0.218** |

| Dataset | Metric | STAN | | | STKGRec | | |
|---|---|---|---|---|---|---|---|
| | | Filt-$d$ | $w/o$ Filt | Ours | Filt-$d$ | $w/o$ Filt | Ours |
| NYC | H@5 | 0.267 | 0.344 | **0.399** | 0.431 | 0.435 | **0.436** |
| | H@10 | 0.343 | 0.406 | **0.461** | 0.514 | 0.518 | **0.523** |
| | G@5 | 0.189 | 0.248 | **0.290** | 0.320 | 0.322 | **0.324** |
| | G@10 | 0.214 | 0.268 | **0.310** | 0.347 | 0.349 | **0.353** |
| TKY | H@5 | 0.200 | 0.282 | **0.310** | 0.420 | 0.426 | **0.427** |
| | H@10 | 0.272 | 0.362 | **0.387** | 0.494 | **0.504** | 0.503 |
| | G@5 | 0.137 | 0.197 | **0.220** | 0.323 | 0.329 | **0.332** |
| | G@10 | 0.160 | 0.223 | **0.245** | 0.347 | 0.354 | **0.356** |

(*i.e.*, $c(u, \bar{v}, t)$) *rather than the distance* (*i.e.*, $d(u, \bar{v}, t)$) is more effective in accurately finding the HN POIs.

Moreover, to verify whether these negative POIs help the model correctly position the user's positive POI in a high ranking during training, we observed the difference in accuracy among three schemes at every epoch during training.

Figures 6 and 7 display the results, where we set the maximum of epochs to 50 and obtained NDCG (G@10) through the validation set for NYC and TKY, respectively. For PLSPL [23] and CatDM [27], Ours considerably contributes to improving the accuracy from the beginning of the training and converges to higher accuracy than the other two schemes. For STAN [15], in the early phase of the training (*i.e.*, less than 5 epochs), the three schemes do not show a significant difference in accuracy. As the training progresses, however, the model trained by Ours shows the greatest improvement in accuracy and finally converges to the highest accuracy. These results indicate that our strategy of filtering out POIs which prioritizes $c(u, \bar{v}, t)$ over $d(u, \bar{v}, t)$ is effective for next-POI recommendations.

*4.2.3 RQ 3.* We designed two variants of our scheme with respect to considering a geographical distance in Step 2: (i) sampling the POIs with the *long distance* from the user (*i.e.*, contrary to our scheme) as HN POIs among the POIs obtained by Step 1 (namely, Dist-$l$)[11]; and (ii) sampling top-$n$ POIs with the highest value of $c(u, \bar{v}, t)$ *without considering the distance* (namely, $w/o$ Dist).

---

[11]In other words, Dist-$l$ finds top-$n$ HN POIs with the highest $DoP_{long}(u, \bar{v}, t)$ by formulating it as $(c(u, \bar{v}, t) \cdot d(u, \bar{v}, t))$.

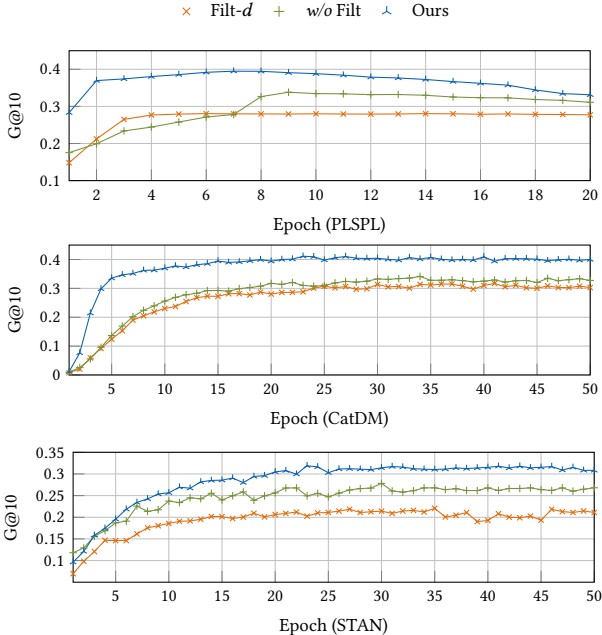

**Figure 6: Accuracy obtained as training proceeds when using PLSPL, CatDM, and STAN as a base model (NYC).**

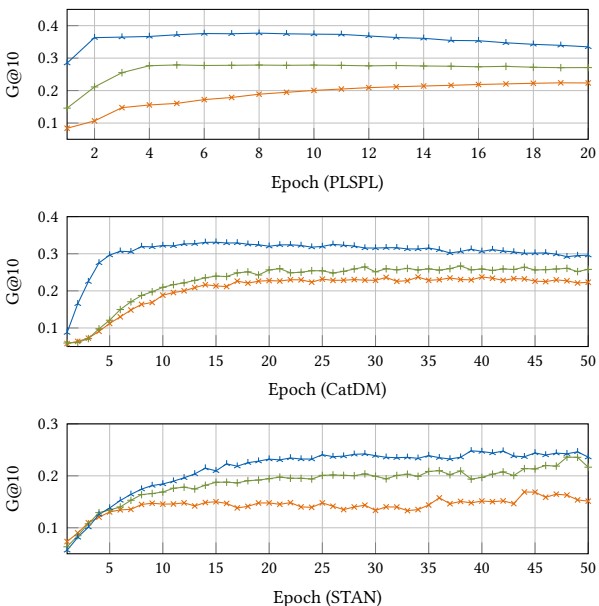

**Figure 7: Accuracy obtained as training proceeds when using PLSPL, CatDM, and STAN as a base model (TKY).**

Table 4 compares the accuracy of Ours and those of the two variants Dist-$l$ and $w/o$ Dist, which exhibits the following observations: (i) exploiting the distance based on the strategy (*i.e.*, Dist-$l$) *opposite* to Ours shows worse accuracy than using only the preference scores to find the HN POIs (*i.e.*, $w/o$ Dist); (ii) pursuing both a high

**Table 4: Comparison of accuracy among our scheme and its two variants with respect to using a distance in Step 2**

| Dataset | Metric | PLSPL | | | CatDM | | |
|---|---|---|---|---|---|---|---|
| | | Dist-$l$ | w/o Dist | Ours | Dist-$l$ | w/o Dist | Ours |
| NYC | H@5 | 0.369 | 0.386 | **0.400** | 0.205 | 0.243 | **0.249** |
| | H@10 | 0.499 | 0.508 | **0.519** | 0.247 | 0.281 | **0.291** |
| | G@5 | 0.253 | 0.265 | **0.281** | 0.168 | 0.211 | **0.216** |
| | G@10 | 0.295 | 0.305 | **0.320** | 0.184 | 0.223 | **0.230** |
| TKY | H@5 | 0.267 | 0.286 | **0.314** | 0.185 | 0.222 | **0.227** |
| | H@10 | 0.401 | 0.406 | **0.433** | 0.238 | 0.272 | **0.282** |
| | G@5 | 0.171 | 0.185 | **0.210** | 0.155 | 0.198 | **0.200** |
| | G@10 | 0.213 | 0.224 | **0.248** | 0.173 | 0.213 | **0.218** |

| Dataset | Metric | STAN | | | STKGRec | | |
|---|---|---|---|---|---|---|---|
| | | Dist-$l$ | w/o Dist | Ours | Dist-$l$ | w/o Dist | Ours |
| NYC | H@5 | 0.367 | 0.381 | **0.399** | 0.418 | 0.425 | **0.436** |
| | H@10 | 0.440 | 0.444 | **0.461** | 0.492 | 0.514 | **0.523** |
| | G@5 | 0.263 | 0.276 | **0.290** | 0.323 | 0.317 | **0.324** |
| | G@10 | 0.287 | 0.297 | **0.310** | 0.347 | 0.347 | **0.353** |
| TKY | H@5 | 0.286 | 0.305 | **0.310** | 0.423 | 0.423 | **0.427** |
| | H@10 | 0.373 | 0.386 | **0.387** | 0.499 | 0.495 | **0.503** |
| | G@5 | 0.203 | 0.212 | **0.220** | 0.326 | 0.329 | **0.332** |
| | G@10 | 0.231 | 0.239 | **0.245** | 0.349 | 0.353 | **0.356** |

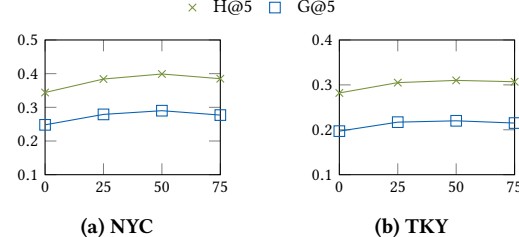

(a) NYC          (b) TKY

**Figure 8: Accuracies obtained by varying $m$ (STAN).**

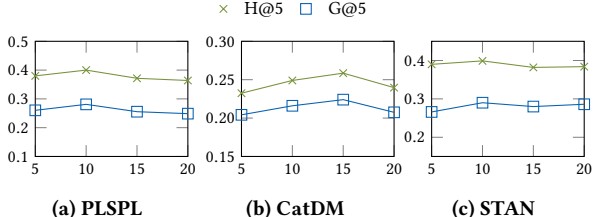

(a) PLSPL          (b) CatDM          (c) STAN

**Figure 9: Accuracies obtained by varying $n$ (NYC).**

preference score and a short distance in HN sampling contributes to enhancing the accuracy.

*4.2.4 RQ 4.* Figure 8 displays the changes in accuracy depending on the number $m$ of candidates for HN POIs from 0 to 75 in increments of 25, where $m$=0 indicates the result of sampling HN POIs without filtering out any negative POIs. Regardless of the value of $m$, we can demonstrate that obtaining candidate HN POIs through filtering is more effective than the case of $m$=0. Additionally, we can observe that the difference in accuracy is *not substantial* based on the value of $m$, and we used 50 as the value for $m$.

Moreover, in Figure 9, we show the changes in accuracy by varying the number $n$ of HN POIs to sample from 5 to 20 in increments of 5.[12] Overall, we can observe that our proposed model-training scheme is *insensitive* to the value of $n$ and we adopted 10 as $n$ which shows marginally better results than the rest.

## 5 RELATED WORK

In general recommendation domains such as OTT and e-commerce, various studies for HN sampling have been conducted. First, DNS [28] and AOBPR [16] regard a negative item with a *high preference score* predicted by the models as *informative* for their model training. The loss over a positive item and the negative item results in *large gradients*, which can help the models be effectively trained in the *correct way*. In particular, AOBPR [16] introduces a way to *efficiently* find HN samples through a *mixture model of the sampling distribution*. In [18], researchers demonstrate that HN sampling is more advantageous than *non-sampling* in optimizing the *One-way Partial AUC* (OPAUC) metric, thereby validating that HN sampling is more effective than non-sampling in improving the top-$N$ recommendation accuracy. Next, [29] claims that sampling items which are hard negative as well as true negative can prevent the model from *overfitting*. In this sense, GDNS [29] presents a way of finding these negative samples by observing changes in predicted preference scores for negative samples over *successive epochs*. Finally, RecNS [26] proposes a HN sampling method for *graph-based recommendation* models (*e.g.*, GNN [6], LightGCN [7]). Our work differs from these studies in that *we propose the model-training scheme based on HN sampling in the next-POI recommendation for the first time*.

## 6 CONCLUSIONS

In this paper, we conducted an *exhaustive and comprehensive* study of the negative sampling issue for the next-POI recommendation, while most existing approaches have simply employed the RN sampling for each user during model training. Based on our key observation that RN sampling performs as *EN sampling* as model training proceeds, we pointed out the limitation of existing studies by validating that EN sampling is more *disadvantageous* than HN sampling in terms of improving accuracy. To address this limitation, we introduced the novel concept of *DoP* that determines HN POIs, proposing the model-training scheme based on HN sampling with the following two steps: (Step 1) filtering out the POIs by the degree of having the characteristics preferred by a user; (Step 2) sampling HN POIs, located close to her positive POI *both in the latent/geographical space*, via DoP. Experimental results demonstrated that all the state-of-the-art models incorporated with our training scheme achieved significant benefits in accuracy.

---

[12]For the results of using another dataset, please refer to Appendix C.

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

# Appendix

Anonymous Author(s)

## ABSTRACT

To recommend the *points of interest* (POIs) that a user would *check-in next*, most *deep-learning* (DL)-based existing studies have employed *random negative* (RN) sampling during model training. In this paper, we claim and validate that, as the training proceeds, such an RN sampling in reality performs as sampling *easy negative* (EN) POIs (*i.e.*, *EN sampling*) that a user was *highly unlikely* to check-in at her check-in time point. Furthermore, we verify that EN sampling is more *disadvantageous* in improving the accuracy than sampling *hard negative* (HN) POIs (*i.e.*, *HN sampling*) that a user was *highly likely* to check-in. To address this limitation, we present the novel concept of the *Degree of Positiveness* (DoP), which can be formulated by two factors: (i) the degree to which a POI has the *characteristics preferred* by a user; (ii) the *geographical distance* between a user and a POI. Then, we propose a new *model-training scheme based on HN sampling* by using DoP. Using real-world datasets (*i.e.*, NYC, TKY, and Brightkite), we demonstrate that all the state-of-the-art models trained by our scheme showed dramatic improvements in accuracy by up to about 82.8%. The code of our proposed scheme is available in an external link (https://anonymous.4open.science/r/code-BF64/).

**ACM Reference Format:**
Anonymous Author(s). 2023. Appendix. In *Proceedings of Make sure to enter the correct conference title from your rights confirmation email (TheWeb-Conf '24)*. ACM, New York, NY, USA, 11 pages. https://doi.org/XXXXXXX. XXXXXXX

## A IMPLEMENTATION DETAIL

### A.1 Environments

Base models for the next-POI recommendation were implemented with PyTorch 1.13.1, Pandas 1.1.5, Tensorflow 1.15.0, scikit-learn 1.0.2, Torchtext 0.6.0, and Python 3.7. All experiments were conducted on desktops with 64 GB memory, Intel i9-10900K CPU (3.7 GHz, 20M cache), and Nvidia GeForce RTX 3070.

### A.2 Dataset Preprocessing

We conducted experiments on three popular real-world datasets, NYC and TKY from Foursquare, and Brightkite. The statistics of these datasets are shown in Table A. For NYC and TKY, which are check-in datasets collected from New York and Tokyo from April 12, 2012, to February 16, 2013, respectively, we kept only users who had visited more than 5 POIs and kept only POIs visited by more than 5 users. For GeoSAN [13], however, we maintained

**Table A: Statistics of three real-world datasets**

| Dataset | # of users | # of POIs | # of check-ins | Sparsity(%) |
|---|---|---|---|---|
| NYC | 1083 | 38,333 | 227,428 | 99.78 |
| TKY | 2293 | 61,858 | 573,703 | 99.85 |
| Brightkite | 51406 | 772,967 | 4,747,287 | 99.99 |

only users whose check-in sequence length was at least 100 due to the memory space issue. For Brightkite, which contains a vast number of check-ins of users from April 2008 to October 2010, we left users whose check-in sequence length was at least 100 and we left POIs visited by more than 10 users. We note that PLSPL [23] and CatDM [27] exploit categories of POIs while training their models, but Brightkite does not include category information. Thus, we obtained category information of POIs from the Foursquare API and then *merged* them into the Brightkite dataset.

### A.3 Evaluation Protocol

To evaluate the accuracy of base models precisely, we divided each dataset into training/validation/test sets and tuned the hyperparameters by following the respective study that proposed each model.

- PLSPL [23]: The training set, validation set, and test set consist of the initial 80%, 10%, and 10% of the check-ins of each user, respectively. We set the batch size, hidden dimension, number of epochs, and learning rate as 32, 128, 20, and 0.001, respectively.
- CatDM [27]: The training set, validation set, and test set consist of the initial 80%, 10%, and 10% of the check-ins of each user, respectively. Within the test set, we selected the last 24-hour check-ins for every user, designating the first POI as the current location, and regarding the subsequent check-ins as the ground truth. We set the batch size, hidden dimension, and learning rate as 1, 64, and 0.001, respectively.
- STAN [15]: The training set, validation set, and test set consist of the initial 80%, 10%, and 10% of the check-ins of each user, respectively. We set the batch size, number of epochs, and learning rate as 1, 50, and 0.003, respectively.
- GeoSAN [13]: For each user's check-in sequence, we regard the last check-in as the test set and the remaining (previous) check-ins as the training set for the user. We divided each user's check-in sequence into subgroups of 100 check-ins. Then, we regarded the last sub-group as the test set and the remaining subgroups as the training set for the user. We set the batch size, hidden dimension, number of epochs, and learning rate as 32, 100, 100, and 0.001, respectively.
- STKGRec [3]: We divided each user's check-in sequence into 24-hour *sessions*, where at least 3 check-ins are contained in each session. The training set, validation set, and test set consist of the initial 80%, 10%, and 10% of the sessions of each user, respectively. We set the batch size, hidden dimension,

### Table B: Notations

| Notation | Description |
|---|---|
| $u$, $v$, and $t$ | A user $u$, a POI $v$, and a time point $t$ |
| $\bar{V}_{u,t}$ | A set of all (negative) POIs that $u$ has not checked-in at $t$ ($\bar{v} \in \bar{V}_{u,t}$) |
| $c(u, v, t)$ | The degree to which $v$ has the characteristics preferred by $u$, given the previous check-in sequence of $u$ up to the time point $t$ |
| $d(u, v, t)$ | The geographical distance between $u$ and $v$ at $t$ |
| $Q_{u,t}$ | The next-POI recommendation model trained by the check-in sequence of $u$ up to $t$ |
| $Q_{u,t}(\bar{v})$ | The preference score of $u$ for $\bar{v}$ predicted by model $Q_{u,t}$ |
| $C_{u,t}(m)$ | A set of $m$ candidates for HN POIs of $u$ at $t$ |
| $DoP(u, \bar{v}, t)$ | DoP of user $u$ for POI $\bar{v}$ at $t$, formulated by $c(u, \bar{v}, t)$ and $d(u, \bar{v}, t)$ |
| $H_{u,t}(n)$ | A set of $n$ HN POIs of $u$ at $t$ |
| $rank(c(u, \bar{v}, t))$ | The ranking of $\bar{v}$ among the POIs in $\bar{V}_{u,t}$ after they are sorted in the descending order of $c(u, \bar{v}, t)$ |
| $rank(DoP(u, \bar{v}, t))$ | The ranking of $\bar{v}$ among the POIs in $C_{u,t}(m)$ after they are sorted in the descending order of $DoP(u, \bar{v}, t)$ |

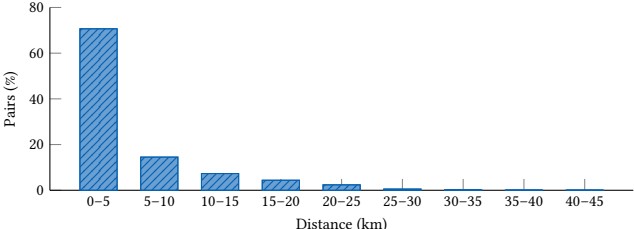

**Figure A: Distribution of pairs of successive POIs over the distance between them (NYC).**

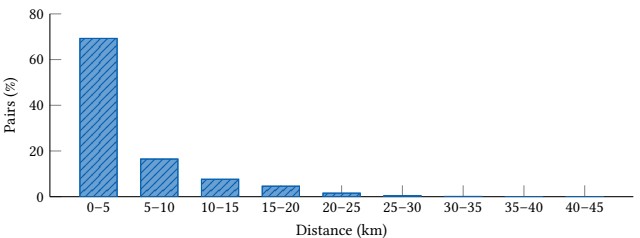

**Figure B: Distribution of pairs of successive POIs over the distance between them (TKY).**

number of epochs, and learning rate as 64, 100, 100, and 0.0001, respectively.

## B  NOTATIONS

Table B summarizes the notations used in this paper.

## C  ADDITIONAL EXPERIMENTAL RESULTS

### C.1  Distribution of pairs of successive POIs over the distance between them

Using real-world datasets, we conducted a statistical analysis to verify the relationship between a user's check-in probability for a POI and the distance between the POI and the user. Figures A and B illustrate the results for the NYC and TKY datasets, respectively.

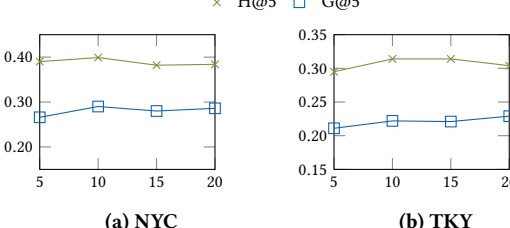

**Figure C: Accuracies obtained by varying $n$ for STAN.**

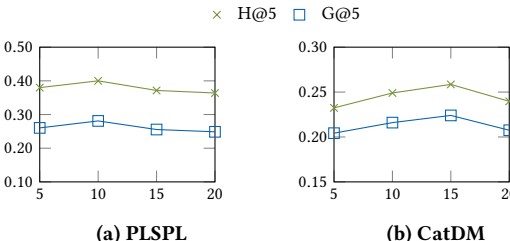

**Figure D: Accuracies obtained by varying $n$ (NYC).**

The $x$-axis indicates the range of a distance and the $y$-axis indicates the ratio of pairs of successive POIs corresponding to the range of a distance. It can be observed that about 70% of all pairs for both datasets belong to the range of a distance less than 5km; users generally have a strong tendency to check-in the POI *located close to their current locations* as their next-POIs.

### C.2  Accuracies obtained by varying $n$

To answer RQ 4, we observe the changes in accuracy by varying the number $n$ of HN POIs to sample from 5 to 20 in increments of 5. First, Figure C shows the results with respect to STAN from using datasets NYC and TKY. In addition, Figure D shows the results for PLSPL and CatDM from using NYC. Overall, we can observe that our proposed model-training scheme is *insensitive* to the value of $n$ and we adopted 10 as $n$ which shows marginally better results than the rest.