# OpenReview forum: "Negative Sampling in Next-POI Recommendations: Observation, Approach, and Evaluation"
_ACM.org/TheWebConf/2024/Conference — TheWebConf24_

### Official Review · Reviewer_1CUZ · 2023-11-16

**Novelty:** 5
**Technical Quality:** 6

**Review:**

This paper claims and validates the disadvantages of random-negative sampling popular in Next-POI recommendations, and proposes a hard-negative-sampling based model training scheme utilizing their proposed Degree of Positiveness. The authors conduct comprehensive experiments to demonstrate its superiority on SOTA models and understand the importance of each component.

Pros:
1. in-depth validation and explanation of the disadvantages of random-negative sampling with plots.
2. novelty in their proposed hard-negative sampling training scheme, with the definition of DoP considering both user preferences and geographical distances.
3. comprehensive experiments with good experiment designs, and promising experiment results.
4. open-sourced code of the proposed scheme.

**Questions:**

1. The authors mention that "multiplication adopted in Eq. 4 has been widely used in other studies for recommender systems [1, 5]". Then what's the novelty here in this paper? How does it compare to the existing works?

**Reviewer Confidence:**

3: The reviewer is confident but not certain that the evaluation is correct

**Scope:**

4: The work is relevant to the Web and to the track, and is of broad interest to the community

---

### Official Review · Reviewer_mNCL · 2023-11-20

**Novelty:** 4
**Technical Quality:** 3

**Review:**

The paper aims to provide a negative sampling method in POI recommendation. The author firstly analyzes the importance of hard negatives and then propose to combine the hardness of negatives and the geographical distance to sample items. However, the simple combination is a little tricky and inefficient. Sampling-based methods are originally designed for efficient training, but the proposed two-step strategy requires the same calculation as using all negative items. And the baselines adopted in the experiments are not convincing.

### Strengths
1. The writing of the paper is fluent and easy to follow.
2. The analysis in section 2 is consistent with results in other negative sampling works and provides an intuitive view to explain why hard negative sampling matters.
3. The idea to combine the hardness of negatives and the geographical distance is indeed an effective method, and the author demonstrates its effectiveness in experiments. Also, geographical distance is one of the key feature in POI recommendations.


### Weaknesses
1. The proposed method is tricky and lacks of novelty. Actually the DoP is just a simple combination of hardness of negative samples and the geographical distance of interests. The two perspectives have been proposed existing works, such as [1][2][3] for hardness and [4] for geographical distance.
2. The method is not efficient and not applicable. In the first step of the method, scores on all items are required to be calculated. The time cost is linear with the number of items. Once all scores are obtained, softmax loss function could be adopted, which would be more effective than the sampling-based methods.
3. The baselines adopted in experiments is not sufficient. Since the main contribution of the paper is the proposed negative sampling method, the baselines should be some competitive sampling methods instead of the random sampling method only, such as DNS[1], SRNS[2], PRIS[3], Kernel-based Sampling[5].

[1] Optimizing top-n collaborative filtering via dynamic negative item sampling.

[2] Simplify and Robustify Negative Sampling for Implicit Collaborative Filtering.

[3] Personalized ranking with importance sampling.

[4] Geography-Aware Sequential Location Recommendation

[5] Adaptive sampled softmax with kernel based sampling.

**Questions:**

1. Can you give analysis on complexity of the proposed method?

2. Since scores of all POIs are calculated, why not use the softmax loss? Usually softmax loss could outperform sampling-based loss but not efficient[1].

3. The baseline methods should be some negative sampling methods instead of various models. The table 1 is actually designed for the model-agnostic analysis.

4. How about use geographical distance only? It’s better to give a detailed analysis on the effect of the hardness and the geographical distance.

5. Since the w/o Dist variant is inferior and the geographical distance is important, why increasing n could not benefit?


[1] Turning Dross Into Gold Loss: is BERT4Rec really better than SASRec?

**Reviewer Confidence:**

4: The reviewer is certain that the evaluation is correct and very familiar with the relevant literature

**Scope:**

4: The work is relevant to the Web and to the track, and is of broad interest to the community

---

### Official Review · Reviewer_2eBP · 2023-11-22

**Novelty:** 4
**Technical Quality:** 3

**Review:**

The paper proposes the concept of Degree of Positiveness (DoP) and a new model-training scheme based on hard negative sampling by using DoP. Experimental results show that this method is able to improve the performance of state-of-the-art models.

Pros:
1.	The paper makes a comprehensive study for the negative sampling issue, which is the first in-depth study of this issue. I think this issue makes a lot of sense.
2.	The proposed method is a generalized method that can be used in all POI recommendation tasks that require negative sampling, and as illustrated in the paper, the improvement of the method is considerable.
3.	The paper provides a new perspective on the study of POI recommendation.

Cons:
1.	The presentation of the paper is confusing. The authors use a lot of italics in the text, but these words are not all technical terms or parts that need to be emphasized. It is difficult for the reader to understand the author's intentions. The writing needs polishing.
2.	The steps of training with DoP can be expressed in pseudo code form, it is hard to follow all the details.
3.	Although, the experiments are comprehensive and convincing, the paper fails to explain how HN sampling influences the training process.

**Questions:**

1.	In Section 2, the authors discuss that RN sampling methods employed in existing studies are close to EN sampling, and conduct experiments to verify this. Is there Is a way to prove this point from a theoretical point of view? It would be more convincing if there were a theory to justify the experimental results.
2.	What is the efficiency of HN sampling using DoP? If possible, add relevant experiments on the training time compared to RN sampling.

**Reviewer Confidence:**

3: The reviewer is confident but not certain that the evaluation is correct

**Scope:**

3: The work is somewhat relevant to the Web and to the track, and is of narrow interest to a sub-community

---

### Official Review · Reviewer_Aqvf · 2023-11-22

**Novelty:** 5
**Technical Quality:** 4

**Review:**

This paper proposes a new model training scheme with hard negative (HN) sampling for next-POI recommendation. The main idea is reasonable that HN sampling considering both user preference and geographical distance could be more beneficial than random negative (RN) sampling. The experiments on multiple datasets demonstrate consistent accuracy improvements over various baseline models.

However, the novelty is incremental on top of existing work on HN sampling for recommendation. The proposed Degree of Positiveness (DoP) measure is conceptually simple by linearly combining two factors. More thorough literature review, comparison, and limitation analysis are needed to better position the contribution.

The paper is overall well-written. More details could be provided on dataset statistics and preprocessing. The experiment protocol and results need to be elaborated for a convincing evaluation.

**Questions:**

1. How does the proposed method compare with the latest HN sampling techniques for recommendation (e.g., AOBPR, DNS, GDNS)? What are the key differences?

2. Have you experimented with more complex formulations of DoP beyond linear combination? How sensitive is the performance to DoP design?

3. Is the experiment adequate? The effect was verified on only 3 datasets, which were limited in size and domain. It is recommended to validate on more data sets with larger scale and wider application areas.

4. Could you provide more details on the dataset statistics and preprocessing steps for reproducibility?

**Reviewer Confidence:**

3: The reviewer is confident but not certain that the evaluation is correct

**Scope:**

3: The work is somewhat relevant to the Web and to the track, and is of narrow interest to a sub-community

---

### Official Review · Reviewer_MFFm · 2023-11-28

**Novelty:** 4
**Technical Quality:** 4

**Review:**

My review are as below:


Strengths:

-	The idea is simple and easy to follow
-	Methods are well explained
-	Various experiments were conducted. According to Table 1, it seems like the proposed method does have improvements.
-	Source code was provided for reproducibility

Weaknesses:

-	For Brightkite dataset with CatDM, why do we see only small gains?
-	In Table 2, all the datasets have similar sparsity. We should test on other datasets with various density
-	This paper has missing ablation studies to convince the readers. Most of the analyses only focus on accuracy.
-	For Figure 6 and 7, it seems like accuracy tends to ‘converge’ for PLSPL and CatDM. Why is that?
-	What is the run time for using the proposed method compared to random sampling?

**Questions:**

Please refer to my comments above.


----------
I've read the rebuttal and increased the technical quality score from 3 to 4

**Reviewer Confidence:**

3: The reviewer is confident but not certain that the evaluation is correct

**Scope:**

3: The work is somewhat relevant to the Web and to the track, and is of narrow interest to a sub-community

---

### Decision · Program_Chairs · 2024-01-22

**Decision:**

Accept

**Comment:**

All reviewers found merits in the submission but some of them also propose a number of concerns. I think the authors have fixed many of these concerns in the rebuttal discussions. Although some reviewers are still not so positive, I believe they will improve their submission per the suggestions.